# Low- vs. High-Power Laser for Holmium Laser Enucleation of Prostate

**DOI:** 10.3390/jcm12052084

**Published:** 2023-03-06

**Authors:** Vasileios Gkolezakis, Bhaskar Kumar Somani, Theodoros Tokas

**Affiliations:** 1Department of Urology, Athens Medical Centre, Distomou 5-7, 151 25 Marousi, Greece; 2Department of Urology, University Hospital Southampton NHS Trust, Southampton SO16 6YD, UK; 3Department of Urology and Andrology, General Hospital Hall i.T., Milser Str. 10., A-6060 Hall in Tirol, Austria; 4Training and Research in Urological Surgery and Technology (T.R.U.S.T.)-Group, Milser Str. 10., A-6060 Hall in Tirol, Austria

**Keywords:** prostate enucleation, laser power, holmium, HoLEP

## Abstract

Holmium laser enucleation of the prostate (HoLEP) constitutes an established technique for treating patients with symptomatic bladder outlet obstruction. Most surgeons perform surgeries using high-power (HP) settings. Nevertheless, HP laser machines are costly, require high-power sockets, and may be linked with increased postoperative dysuria. Low-power (LP) lasers could overcome these drawbacks without compromising postoperative outcomes. Nevertheless, there is a paucity of data regarding LP laser settings during HoLEP, as most endourologists are hesitant to apply them in their clinical practice. We aimed to provide an up-to-date narrative looking at the impact of LP settings in HoLEP and comparing LP with HP HoLEP. According to current evidence, intra- and post-operative outcomes as well as complication rates are independent of the laser power level. LP HoLEP is feasible, safe, and effective and may improve postoperative irritative and storage symptoms.

## 1. Introduction

Benign prostate hyperplasia (BPH) with consecutive lower urinary tract symptoms (LUTS) constitutes a significant health issue of the aging male. Traditional transurethral resection of the prostate (TURP) remains the standard treatment for small- and medium-sized prostate glands and patients who fail medical therapy and may have complications of outlet obstruction such as bladder stones, urinary retention, or renal insufficiency [1]. Nonetheless, between several surgical treatment modalities, transurethral holmium laser enucleation of the prostate (HoLEP) has emerged as a distinctive technique that can be applied to prostates of all sizes [2,3,4]. Compared to standard TURP, HoLEP offers better hemostasis, shorter catheterization and hospitalization times, and nullifies the rate of TURP syndrome [3,5]. The holmium laser technology enables the prostatic tissue to be enucleated from the capsule while simultaneously coagulating the capsular surface. HoLEP, which offers long-term functional results superior to TURP and comparable to open simple prostatectomy but with lower treatment morbidity and complication rates, is therefore regarded as a procedure of reference for the surgical treatment of large prostate glands [2,3,4]. Surgeons usually apply power settings of 80–100 W with 2 J energy and 40–50 Hz frequency and an occasional power reduction for coagulation (75 W, 1.5 J, and 50 Hz) and apical preparation (30 W, 0.6 J, and 50 Hz) [6,7]. These settings provide the ability to adjust pulse duration to energy and frequency but also necessitate more expensive equipment with numerous high-power plugs, which are generally considered limitations of the widespread adoption of HoLEP.

Low-power (LP) devices are also available on the market, functioning at powers of 20, 30, and 50 W with lower startup costs and no demand for specialized plugs. The same equipment can be used successfully for lithotripsy and BPH surgery. Comparing these qualities to high-power (HP) units could be advantageous. Rassweiler et al. were the first to use LP settings (24 W, 2 J, and 12 Hz, or 39.6 W, 2.2 J, and 18 Hz) to treat 129 patients, proving the treatment’s viability, safety, and effectiveness [7]. These findings implied a significant reduction in the initial capital equipment cost, which may make adopting this technique more bearable if the method’s effectiveness is preserved. Despite this, the LP method has not gained much support, and there are still few reports on LP HoLEP in the era of rising HP machine output. On the other hand, the higher price tag that comes with these sophisticated devices is a significant disadvantage, and many endourologists are looking for less expensive options. Furthermore, while HP and unique technologies like MOSES may only be available in referral centers, LP holmium laser machines are universally available, as they are frequently adopted for other endourological procedures (i.e., lithotripsy). Hence, this work aimed to compare LP to standard HP HoLEP regarding perioperative parameters, complications, and functional outcomes.

## 2. Materials and Methods

### 2.1. Literature Search

We performed a literature search in PubMed and used the following keywords: ‘prostat* hypertrophy’, ‘prostat* hyperplasia’, ‘BPH’, ‘BPO’, ‘HoLEP’, and ‘holmium laser’. We limited our search to papers written in English.

### 2.2. Selection Criteria

The PICOS (patient, intervention, comparison, outcome, study type) model was employed. Patient: adults undergoing HoLEP for BPH; intervention: LP HoLEP; comparison: HP HoLEP; outcome: surgical time, operative efficiency, postoperative catheterization time, length of hospital stay, re-catheterization, blood transfusion, incontinence rate, international prostate symptom score (IPSS), maximum peak flow (Qmax), and post-void residual urine (PVR) at last follow-up; study type: randomized, prospective non-randomized, and retrospective studies. HP HoLEP was conducted at 100 W, whereas LP was performed at 30 W, 40 W, and 50 W. Due to study heterogeneity and the non-standardized quality appraisal, we performed a narrative synthesis. The limitations of using a single database for a review are also taken into account [8]. Furthermore, our results might be constrained by study heterogeneity and selection bias. We included randomized, prospective non-randomized, and retrospective studies and excluded case studies, reviews, and editorials.

## 3. Results

The literature search identified 969 records (Figure 1). Additionally, we included four studies from other sources. Following title, abstract, and full-text screening, we selected and included 11 studies in the review (Table 1 and Table 2). Five studies described the functions and presented outcomes of LP HoLEP and six compared LP with HP HoLEP in terms of procedure times, peri- and postoperative outcomes, and complications. We included four meeting abstracts [9,10,11,12], one prospective comparative study [13], one prospective randomized trial [14], one prospective case series [15], three retrospective case series [16,17,18] and one ex vivo porcine study [19].

### 3.1. Efficiency and Speed of LP HoLEP

Operative time (OT), enucleation time (ET), operative efficiency (OE; defined as resected prostate weight divided by operative time in g/min), enucleation efficiency (EE; defined as resected prostate weight divided by enucleation time in g/min), laser/prostate ratio (defined as laser energy consumed divided by resected prostate weight, in KJ/g), and laser rate (defined as the laser energy consumed) were among the outcome measures evaluated. In the first randomized controlled trial comparing LP (50 W, 2 J, and 25 Hz) versus traditional HoLEP (100 W, 2 J, and 50 Hz), the authors found comparable outcomes in terms of EE (1.42 ± 0.6 g/min vs. 1.47 ± 0.6 g/min), OE (1.01 ± 0.4 g/min vs. 1.09 ± 0.4 g/min), and OT (81 min vs. 75.5 min) regardless of the surgeon’s experience [14]. Two prospective case-control studies from the same group presenting the records of 316 patients with any prostate volume (range 10–200 g), normal PSA, Qmax < 15 mL/s, IPSS > 10, and PVR < 300 cc and comparing the efficacy of en-bloc no-touch LP HoLEP (40 W, 2 J, and 20 Hz) with HP HoLEP (100 W, 2 J, and 50 Hz) revealed identical results regarding mean ET (27.5 min vs. 31 min) and EE (1.7 g/min vs. 1.64 g/min, respectively) in the hands of an experienced surgeon [10,11]. The authors reported a reduction in energy consumption of nearly one-third.

Gazel et al. compared the impact of two different LP settings on enucleation and hemostasis in 160 patients and recorded increased EE (1.2 vs. 0.78 g/min, *p* = 0.001) while administering 37.5 W (1.5 J and 25 Hz) as opposed to 20 W (1 J and 20 Hz) [17]. In addition, the mean enucleation rate (0.64 vs. 0.88%, *p* = 0.001) and laser efficiency (2.07 vs. 2.12 joule/g, *p* = 0.003) were significantly higher with 37.5 W. The enucleation time was significantly shorter (54 vs. 75.5 min, *p* = 0.002). The authors concluded that using 37.5 W, both enucleation and hemostasis could be performed successfully, while using 100 W in the bladder neck shortens the duration of the procedure. Furthermore, in an experimental ex vivo study, Yilmaz et al. demonstrated that the HP–Ho:YAG’s efficiency (evaluated by a numerical measurement of the “tissue pocket” created by separating the fascial layers of a porcine belly, measured in cm^2^/min) was reduced by 50% in an LP (3 J and 10 Hz) compared to a medium-power (3 J, 25 Hz) laser setting. Additionally, the authors demonstrated more favorable dissection results with HP systems applying high single-pulse energy, short pulses, and medium frequency [19].

### 3.2. Functional Outcomes of LP HoLEP

Two prospective studies showed significant improvement in IPSS scores and Qmax at three months (24 vs. 5, *p* < 0.001 and 7.8 mL/s vs. 28 mL/s, *p* < 0.001, respectively) [13] and 12 months follow-up (22 vs. 6, *p* < 0.001 and 12 mL/s vs. 29.3 mL/s, *p* < 0.001, respectively) [14] compared to the preoperative assessment. The authors also observed significant improvements regarding PVR at three months (100 mL vs. 30 mL, *p* < 0.001) and 12 months follow-up (135 mL vs. 11.15 mL, *p* < 0.001) [15]. Gilling et al. prospectively compared LP to HP HoLEP and observed a considerable and persistent improvement in these parameters for both energies at up to 12 months follow-up compared to baseline (30.9 mL/s vs. 7.7 mL/s for 50 W setting, 19 mL/s vs. 7.4 mL/s for 100 W setting, statistical evaluation not reported) [9]. In addition, one prospective comparative study [10] and one randomized trial [14] showed similar results with no difference among LP and HP HoLEP regarding IPSS scores at three months follow-up (6.5 ± 5 d.s. vs. 7.8 ± 5 d.s.) [10] and IPSS scores and Qmax at 12 months follow-up (3 vs. 4, *p* = 0.4, 21.1 mL/s vs. 21.8 mL/s, *p* = 0.7) [14].

### 3.3. Postoperative Stress Urinary Incontinence (SIU) and Dysuria after LP HoLEP

Becker et al. reported postoperative SIU rates of 16.7% after one month, declining to 0% after six months [15], whereas Gazel et al. showed similar postoperative SIU rates with both 20 W and 37.5 W energy settings (3.7 vs. 2.5% at three months and 1.2 vs. 0% at 12 months follow-up) [17]. Minagawa et al. assessed SIU retrospectively and found that in 55 patients without preoperative SIU, postoperative SIU was observed in seven patients (12.7%) at one month postoperatively and in three patients (5.5%) at three months postoperatively [18]. Scoffone et al. [10] and Elshal et al. [14] demonstrated similar long-lasting incontinence rates at three months (1.6 vs. 1.4%) [10] and four months (1.6 vs. 1.7%) [14] in patients undergoing LP HoLEP and HP HoLEP.

### 3.4. Safety of LP HoLEP

Reported complication rates of LP HoLEP ranged from 7 to 24% [12]. Of them, 3.7% were Clavien grade 3a, and 5.5% were Clavien 3b [15]. Transfusion rates varied from 0%, with only one case among 74 patients requiring hemostasis under anesthesia (1.3% Clavien grade 3a) [18], to 5% in large adenomas >80 g [12]. The hemoglobin decrease typically varies from 0.5 g/dL [17] to 1.5 g/dL [15]. Tokatli et al. found that patients who had undergone prostate biopsy before HoLEP treatment had a significant hemoglobin drop (*p* = 0.002) regardless of the type of laser device used [13]. The excellent coagulation effect obtained with LP HoLEP was confirmed in the randomized controlled trial by Elshal et al.; the authors found no statistically significant difference between LP and HP HoLEP in median perioperative hemoglobin deficit (0.9 vs. 0.7, *p* = 0.6), blood transfusion rate (0% vs. 0%), median hospital stay (1 day vs. 1 day, *p* = 0.052) and time to catheter removal (1 day vs. 1 day, *p* = 0.7) [14]. Occasionally, LP HoLEP devices produced results that were marginally preferable, such as shorter median catheter times (17.5 vs. 25.1 h) and recovery times (26.6 vs. 32.5 h), although statistical significance was not reached in these cases [9]. When comparing the mean catheterization time (42 h for the 20 W setting vs. 27 h for the 37.5 W setting, *p* = 0.008), Gazel et al. recorded a significant improvement with 37.5 W, whereas no significant difference was found in terms of mean hospitalization time (28 vs. 33 h, *p* = 0.16) [17]. Furthermore, in two prospective and one retrospective LP case series, the median time to catheter removal was 2 [13,15] and 2.6 days [18]. The median hospital stay ranged from 2 [15] to 5.3 days [18]. A statistically significant shorter length of stay was observed in patients with a previous transperineal biopsy (1.3 vs. 3 days, *p* < 0.001) [16].

## 4. Opinion

Since the initial groundbreaking research [20,21], the HoLEP treatment has advanced alongside other developments in urological technology [2,3,4]. According to the most recent research, one pedal should deliver a high laser intensity (>80 W) throughout the entire procedure [18,22,23]. While a quick enucleation can be achieved this way, irritative symptoms frequently remain even a year later [18]. In a recent editorial, Scoffone et al. [23] reported that they retained enucleation effectiveness and efficiency while reducing the laser photothermic effect on the capsule by decreasing the power output from 50 to 20 W. Several authors backed this finding by showing that LP is just as effective as HP HoLEP [12,15,18]. Cecchetti et al.’s “in-vitro” research [24] convincingly showed how different holmium laser settings interact with shockwaves and produce temperatures on soft tissues. The researchers demonstrated that the lowest threshold for plasma bubble generation and shockwave noise for soft tissue ablation was detected at an energy level of 1.4 J and a frequency of 10 Hz. With a particular quantity of joules provided at a lower frequency and an additional longer pulse duration, thermal relaxation time is significantly increased, fewer photothermic side effects are created, and the photomechanical effects are softened while also maintaining laser effectiveness [24,25]. However, it is crucial to remember that while the frequency can be dropped somewhat proportionately, the energy should not be decreased significantly [26].

It is challenging to predict how a laser will affect a particular type of tissue because of the complex interactions between the laser (wavelength, absorption coefficient, power, and pulse), tissue (water concentration, hardness, and absorption coefficient), environment (air and liquid), and the distance and inclination angle between the fibre tip and tissue [27]. A vaporization zone (vaporization volume), incision depth, width, coagulation zone, carbonization zone, and thermo-mechanical or laser damage zone are the parameters that define laser incisions, which are created by explosive tissue water vaporization [28]. Protein denaturation and pyrolysis induce thermal coagulation to develop between 60 and 100 °C. The release of carbon atoms after the vaporization of water molecules causes the adjacent tissue to become carbonized [29]. Perfusion simplifies the process of transferring heat from the laser incision into healthy tissue below, reducing heat damage. However, perfused and non-perfused porcine kidneys show similar laser damage zones [30]. The type of laser used during endoscopic prostate enucleation can affect the type of incision and the power settings used [31]. Using HP lasers, faster procedure times and more significant hemostasis can be achieved, along with broader and deeper tissue incisions [32]. However, when operating near the prostate pseudo-capsule, deeper incisions, especially with a more expansive thermo-mechanical damage zone, may result in collateral damage, such as a neurovascular bundle injury [4]. In contrast, the minimal carbonization zone associated with LP lasers might reduce postoperative urge symptoms [33] and improve histological findings [28].

The most crucial distinction between LP and HP HoLEP is operational effectiveness. The primary evidence for the efficiency of LP HoLEP was provided by the single-series retrospective investigation by Minagawa et al. [18]. In this study, HoLEP procedures were carried out using an 80 W device with a 30 W power setting by surgeons with various surgical skills. HoLEP was successfully treated on every occasion, regardless of the LP setting, without increasing the laser’s output, and no patient required a blood transfusion. Furthermore, the authors assessed the outcomes while considering the surgeon’s level of expertise and concluded that the enucleation time was significantly reduced when an experienced surgeon carried out the HoLEP operation. Moreover, the EE results aligned with other publications that used an HP laser.

The level of surgical experience may be a significant confounder affecting the procedure results. Without utilizing a control group, a study looked at the EE of HoLEP surgery performed by two experienced surgeons using a 50 W device (2.2 J and 18 Hz) [15]. The authors found that their EE values were higher than those reached by HP laser devices after comparing their results to those of earlier HP HoLEP series. They also stressed that the surgeon’s experience is more crucial than the device’s power for acquiring high EE values. Elshal et al. [14] observed no statistically significant differences between the two groups for any operational parameters, including EE values, in the first randomized controlled experiment contrasting LP HoLEP vs. conventional HP HoLEP (50 W and 100 W energy settings). When contrasting the findings of studies comparing the effects of various energy settings on the efficiency of enucleation during the HoLEP procedure, it is apparent that EE values were reported to be lower in studies conducted before 2013 (mean EE values ranged between 0.45 and 0.94 g/min) [7,34] compared with studies conducted in 2017 or later (mean EE values ranged between 1.1 and 1.7 g/min) [14,15]. These findings demonstrate the importance of gaining experience over time by showing that regardless of the device’s power, the procedure’s effectiveness improves as the surgeon’s experience increases. This conclusion implies that starting HoLEP with large-volume prostates is not advisable for novice surgeons.

Endourologists commonly base their selections on the safety of the procedure and the potential for excessive intraoperative bleeding when selecting the laser settings for HoLEP. Using 25 W vs. 40 W power settings in the initial experiment, Rassweiler et al. [7] discovered an average hemoglobin reduction of 3.1 g/dL and an 8% transfusion rate in their patient groups. Despite the unexpectedly high results, several papers with patients receiving procedures with LP settings have reported acceptable values. In their prospective investigation, Becker et al. [15] used the 39.6 W energy setting on 50 W Ho:YAG laser equipment for HoLEP surgery and collected data on more than 50 patients. They noted a 1.9% transfusion rate and an average hemoglobin decrease of 1.5 g/dL. In a different study comparing 50 W and 100 W energy settings, the decrease in hemoglobin was 0.9 and 0.7 g/dL, respectively [18]. Results from a multiple regression analysis by Tokatli et al. [13] revealed that the sole independent predictor of hemoglobin decline was the existence of biopsy anamnesis. Some underlying factors for the increased bleeding risk include acute or chronic inflammatory reactions that cause granulation in the tissue. HoLEP and removing the adenomatous tissue from the prostatic capsule may also be more challenging to accomplish if there is an inflammatory reaction following the biopsy.

One of the problematic consequences of the HoLEP procedure is postoperative SIU, which can occur to an extent in between 2–15% of patients [35,36,37]. SIU is typically described as temporary, which is reassuring the patients and adds significantly to the preoperative counseling process [38]. The two leading causes of SIU are significant urethral sphincter traction during surgery and tissue damage caused by laser energy close to the prostate’s apex. The likelihood of SIU can be decreased by the adenoma’s low energy consumption close to the urethral sphincter. Prospectively, Becker et al. [15] found that the postoperative immediate SIU rate at 1-month follow-up was high (16.7%). However, the rate of SIU decreased to zero by the 6-month follow-up, in line with what is seen with other HP HoLEP series, TURP, and open prostatectomy [3,36,39,40,41].

Research teams using LP HoLEP report results equivalent to those seen when using HP settings. However, the fact that every surgeon who reports on LP HoLEP uses a different enucleation technique, adding the advantages of each to the LP settings, may be a source of bias. We could also converse concerning how to interpret each outcome measure. For example, the length of stay in the hospital is another indirect indicator that may be highly “environment-dependent” since a surgeon may be reluctant to remove the catheter too soon for fear that doing so will result in an immediate re-catheterization (e.g., hospital stay time is longer in Japan for this reason due to the insurance system). Even enucleation efficiency, which seems to be a very reliable indicator of the intraoperative outcome and reflects both the efficiency of the laser and the clarity of vision in a particular setting, may be influenced by several factors, such as the size of the adenoma (large adenomas significantly improve it, while smaller ones worsen it), so the range of adenoma volume within a case series may influence this factor. Also, the narrative structure of this work underscores the paucity of reliable evidence on this subject. The works included in this study are heterogeneous, particularly regarding the types of laser fibres and the laser and irrigation settings used. In addition, the descriptions of tissue effects during laser ablation differ between study groups in terms of definitions, units, and extra details like laser activation/deactivation intervals or laser tip/tissue distance. The critical endpoints and objectives of the included papers also vary. Further multicentric studies are needed to determine how variables deemed necessary for enucleation, such as prostate size, surgical technique, and surgeon expertise, may alter the outcomes of problems when different holmium machines are used [42].

## 5. Conclusions

LP HoLEP is feasible, safe, and effective and may help lessen the frequency, severity, and duration of postoperative dysuria and storage symptoms. The laser power level does not significantly affect the intra- and postoperative variables and the complication rates. While more comparative studies are still required to confirm the efficacy of LP HoLEP with various enucleation techniques, the physical background for LP HoLEP is valid and supports its use, encouraging surgeons with access to LP machines to use this method.

## Figures and Tables

**Figure 1 jcm-12-02084-f001:**
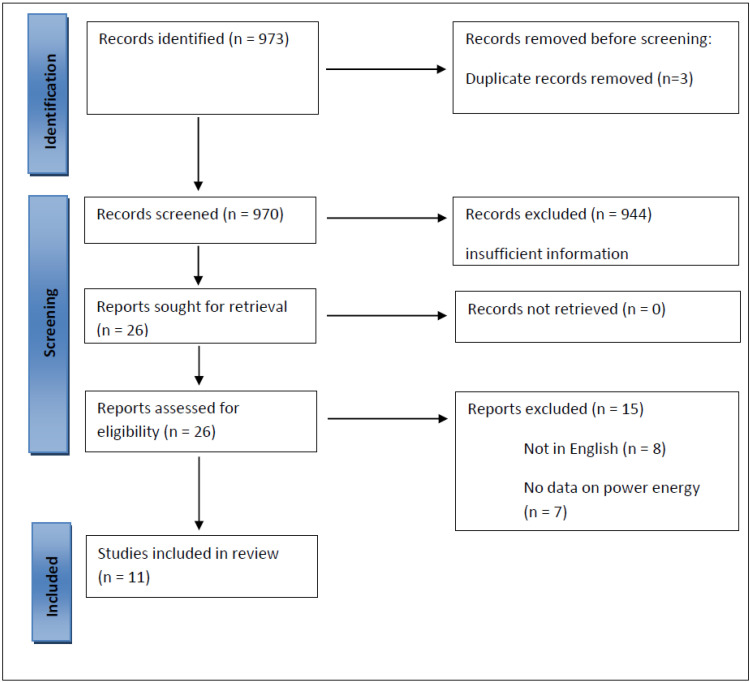
Flow chart.

**Table 1 jcm-12-02084-t001:** Characteristics of low-power HoLEP studies with initial efficacy parameter.

Author	Design	Laser	Pts (N)	Laser Power	Type of Enucleation	Mean Age ± SD (Years)	Mean PV ± SD (mL)	Mean Adenoma Weight (gr)	Median Time to Catheter Removal (Days) (Range)	Median Hospital Stay (Days)
Becker 2018 [15]	Prospective, not randomized, full text	LP HoLEP	54	2.2 J, 18 Hz~39.6 W	Modified technique	72.5	74.5	46	2	2
Bell 2019 [16]	Retrospective, full text	LP HoLEP (with previous TBP) LP HoLEP (biopsy-naïve)	24 85	50 W 50 W	3-lobe 3-lobe	66.8 ± 8.2 71.8 ± 8.7	76.1 ± 35.0 69.3 ± 31.8	NR NR	NR NR	1.3 3.0
Gazel 2020 [17]	Retrospective, full text	LP HoLEP LP HoLEP	80 80	1 J, 20 Hz~20 W 1.5 J, 25 Hz~37.5 W	3-lobe 3-lobe	63 ± 7.97 62 ± 7.07	79 ± 35.71 68.5 ± 47.37	NR NR	42 ± 27.74 (hours) 27 ± 14.38 (hours)	28 ± 6.06 (hours) 33 ± 8.03 (hours)
Minagawa 2017 [18]	Retrospective, full text	HP HoLEP	74	1.5 J, 20 Hz~30 W	En-bloc	75.4 ± 7.1	94.5 ± 61.0	51.8	2.6	5.3
Tokatli 2021 [13]	Prospective, randomized, full text	MP HoLEP MP HoLEP	60 60	2.2 J, 18 Hz~39.6 W 1.2 J, 35 Hz~42 W	2-lobe and 3-lobe 2-lobe and 3-lobe	66.5 67	95 91	44 40	2 2	3 3

**Table 2 jcm-12-02084-t002:** Characteristics of included studies comparing low-power vs. high-power holmium laser enucleation of the prostate with initial efficacy parameter.

Author	Design	Laser	Pts (N)	Laser Power	Type of Enucleation	Mean Age ± SD (Years)	Mean PV ± SD (mL)	Mean Adenoma Weight (gr)	Median Time to Catheter Removal (Days) (Range)	Median Hospital Stay (Days)
Gilling 2013 [9]	Prospective, meeting abstract	LP HoLEP HP HoLEP	20 20	50 W 100 W	NR NR	67.4 ± 11.2 68.9 ± 2.0	NR N	22.4 21.7	17.5 (hours) 25.1 (hours)	26.6 (hours) 32.5 (hours)
Scoffone 2017 [10]	Prospective, meeting abstract	LP HoLEP HP HoLEP	102 214	2.2 J, 18 Hz~40 W 2 J, 50 Hz~100 W	En-bloc En-bloc	67.7 ± 8 69.4 ± 7.5	NR NR	46 55	NR NR	NR NR
Cracco 2017 [11]	Prospective, meeting abstract	LP HoLEP HP HoLEP	102 214	2.2 J, 18 Hz~40 W 52 J, 50 Hz~100 W	En-bloc En-bloc	67.7 ± 8 69.4 ± 7.5	NR NR	46 55	NR NR	NR NR
Elshal 2018 [14]	Prospective, randomized controlled trial, full text	LP HoLEP HP HoLEP	61 60	2 J, 25 Hz~50 W 2 J, 50 Hz~100 W	2-lobe and 3-lobe 2-lobe and 3-lobe	66.4 ± 7 67.0 ± 7	137.6 ± 58 137.6 ± 58	75.5 77	1 (1–5) 1 (1–5)	1 (1–3) 1 (1–5)
Cracco 2020 [12]	Retrospective, meeting abstract	LP HoLEP HP HoLEP	326 212	40 W 100 W	Partial and total en-bloc Partial en-bloc	NR NR	NR NR	48,9 53.3	NR NR	NR NR
Yilmaz 2022 [19]	Ex vivo, porcine belly	LP HoLEP HP HoLEP	NR NR	3.5 J, 10 Hz~35 W 4.5 J, 22.2 Hz~100 W	NR NR	NR NR	NR NR	NR NR	NR NR	NR NR

## Data Availability

Not applicable.

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
