# Peer review of "Low- vs. High-Power Laser for Holmium Laser Enucleation of Prostate"

_jcm, 2023, doi:10.3390/jcm12052084_

Round 1
Reviewer 1 Report
I want to congratulate for the great review about low and high power laser enucleation. I would like to a brief description of equipment available for low and high power.
This review article summarizes evidence in low power vs high power.
Totally relevant. I was lucky to be on residency program that compare low and high power. We have áuriga and lumenis equipment. Many centers do not have all these resources. They never trained on this standard procedure, due to false statement high power is indispensable for holep. I think publication like this filled the gap of knowledge.
In my opinion low power always impact on coagulation capabilities. But there is no current evidence to support this statement.
All references are appropriate.
Table could be improved considering equipment used and hb pre and postoperative. I understand this is heterogeneous samples.
I want to congratulate to authors for this great paper.
Author Response
Dear reviewer,
Thank you for your comments. We have added additional information regarding pre- and postoperative Hb difference. However, it was not possible to include this information in the tables due to size restrictions. Additionally, Hb decrease is not a main factor as only a few studies deal with it.
Reviewer 2 Report
A well-designed original study that aims to show that low-power lasers also has good results in prostatic enucleation, which helps in the democratization of the technique. Congratulations for the work in this paper.
The study loses strength when reporting the results in a narrative manner and due to the high level of heterogeneity between the studies.
Conclusions are consistent with the evidence presented and addressed the main question posed.
Table are ok.
Author Response
Dear reviewer,
Thank you for your comments. We understand the limitations of the study. Some of those is the heterogeneity and low quality of including studies. We have updated the limitation part of the opininion/discussion.
Reviewer 3 Report
It's an interesting review.
I ask if it is possible to write the percentage of the use in the world of LP and HP HOLEP.
In the abstract does not appear (LP) in brackets
Author Response
Dear reviewer,
Thank you for your comments.
We included the abbreviation (LP) in the abstract. It is extremely difficult to include the percentages of LP vs HP worldwide as there are no studies dealing with this subject.
Reviewer 4 Report
The subject of this study is to examine the difference between LP and HP in hoLEP. I think it's an interesting topic. But there are some doubts. First, among the studies analyzed in this study, there is only one prospective, randomized, full test study. In my opinion, looking at the results of the study by Elshal 2018 [14], there seems to be no difference in the surgical methods of LP and HP. Second, the number of participants in this study (Elshal 2018 [14] ) is very small. Third, the analysis including Elshal 2018 [14] in this study is only . Functional outcomes of LP HoLEP are only part, and in other results, this Elshal 2018 [14] is not included. It is considered that the evidence power of the study results is too low.
Author Response
Dear reviewer,
Thank you for your comments. We tried to re-structure the presented data of Elshal study. Additionally, we added a discussion with a limitation part including all reviewer suggestions.
Round 2
Reviewer 4 Report
I think you have corrected my queries well.